# A Blockchain Approach for Migrating a Cyber-Physical Water Monitoring Solution to a Decentralized Architecture

**Bogdan-Ionut Pahontu** [1], **Adrian Petcu** [2], **Alexandru Predescu** [1,*], **Diana Andreea Arsene** [1,*] **and Mariana Mocanu** [1]

[1] Computer Science Department, National University of Science and Technology Politehnica Bucharest, 060042 Bucharest, Romania; bogdan.pahontu@upb.ro (B.-I.P.); mariana.mocanu@upb.ro (M.M.)

[2] Department of Telecommunications, National University of Science and Technology Politehnica Bucharest, 060042 Bucharest, Romania; adrian.petcu@stud.etti.upb.ro

[*] Correspondence: alexandru.predescu@upb.ro (A.P.); diana.arsene@upb.ro (D.A.A.)

**Abstract:** Water is one of the most important resources in our lives, and because of this, the interest in water management systems is growing constantly. A primary concern regarding urban water distribution is how to build robust solutions to facilitate water monitoring flows with the support of consumer involvement. Crowdsensing solutions contribute to the involvement in social platforms for increased awareness about the importance of water resources based on incentives and rewards. Blockchain is one of the technologies that has become increasingly popular in the last few years. The possibility of using this architecture in such different sectors while integrating emerging concepts, such as crowdsensing, the Internet of Things, serious gaming, and decision support systems, offers a lot of alternatives and approaches for designing modern applications. This paper aims to present how these technologies can be combined in order to migrate the functionalities of a water distribution management system from a centralized architecture to a decentralized one by leveraging blockchain technologies. The proposed application was designed to facilitate incident reporting flows in public water distribution networks. The proposed solution was to migrate the rewarding mechanisms using the Ethereum infrastructure. The novelty of this solution is determined by the introduction of this decentralized approach into the architecture and also by increasing customer interest by offering tradeable rewards and dynamic subscription discounts. This results in a new decentralized architecture that allows for more transparent interactions between the water provider and clients and increases customer engagement to contribute to water reporting flows.

**Keywords:** blockchain; water management systems; cyber-physical systems; serious gaming; Internet of Things; crowdsensing solutions

## 1. Introduction

Water management is one of the topics that has attracted a high level of interest in the last few years. Considering this, a lot of proposals have appeared within the scope of optimizing resource consumption and management, as well as educating the end users to better handle their consumption. While many technologies have been proposed to improve water administration processes, there are still concerns about the best methods to increase awareness of the importance of this resource as well as overall consumer involvement in water management flows. Furthermore, there are also projects that focus on monitoring and maintaining the health of the aquatic ecosystem (nutrients and various invasive species). With the help of remote water monitoring, prevention, and warning systems, the large amounts of data collected can serve as decision making support for consumers and suppliers, within the scope of making decisions and communicating them to the public in a transparent way.

As technology advances, people gradually adapt to the multiple transformations that appear, whether the domain is home automation, energy efficiency, entertainment, or health

applications. Both consumers and suppliers benefit from the advantages of intelligent systems in the context of cyber-physical systems. CPSs have already achieved substantial progress in several fields of activity, e.g., robotics, communications, and security. The Internet of Things is also part of the digital transformation, and the collection, processing, and analysis of real-time data using intelligent sensors stimulates the development of intelligent CPS solutions [1]. Crowdsensing has emerged as a promising alternative for addressing complex social and environmental challenges, leveraging the power of citizen engagement and mobile technologies [2]. Crowdsensing platforms have been employed to engage citizens in water quality monitoring initiatives, enabling the collection of data on various parameters [3]. From a wider perspective, these platforms leverage mobile devices and user-contributed data to improve data quality for various environmental monitoring scenarios [4]. Furthermore, the application of citizen knowledge and serious gaming in the context of smart solutions for water monitoring using blockchain architectures has gained significant attention from researchers and practitioners. Several research efforts have explored the potential of blockchain in water monitoring, demonstrating its ability to enhance data reliability, streamline data exchange among stakeholders, and enable automated smart contracts for water management transactions. Serious gaming in water management has shown promising results in improving decision-making processes, enhancing stakeholder engagement, and facilitating active learning [5].

While serious gaming, blockchain, and crowdsensing have been individually explored in the context of water monitoring, their integration holds immense potential. The aim of the research in this paper is to study the state of the existing architectures implemented in the water management field with support for blockchain technologies. The proposed solution in this paper aims to combine emerging technologies in the context of a water monitoring solution to consolidate collaborative monitoring efforts, empower citizens to participate in water management processes, and facilitate data sharing among stakeholders. The solution was designed on Ethereum blockchain and consists of implementing a decentralized marketplace in which clients will be able to purchase discounts for their water subscription contracts. Moreover, the solution facilitates buying and selling both purchased discounts and crypto-tokens received inside the crowdsensing application.

The research novelty is represented by the integrated solution and the scenarios available for end users to interact in a decentralized architecture, obtain monetizable rewards, and trade crypto-tokens for digital water subscription discounts. The following describes how the paper is structured: Section 2 presents related work through a discussion of relevant studies, with a focus on the effects of cyber-physical systems, blockchain, serious gaming, and crowdsensing solutions for water resource management. In Section 3, the theoretical context is described and the methodology for the proposed solution is outlined. In Section 4, the main components are presented in the context of the baseline solution which is the subject of the proposed migration. Section 5 presents the materials and methods proposed and the method of migrating from the traditional approach to a decentralized one. Section 6 presents the actual implementation with the challenges and solutions that were found, and last but not least, Section 7 outlines the conclusion of our work and the future steps that are planned to be performed.

## 2. Related Work

Modern research fields in the context of cyber-physical systems (CPSs) are becoming increasingly popular, while CPSs have become indispensable in the design of future engineering systems. The technological development in recent decades has had a strong impact day by day, making important progress in the development of cyber-physical systems and applications.

CPS applications in the context of water resource management promote water sustainability while increasing efficiency, automation, and consumer confidence. One of the most important applications of CPSs in the field of water management is the monitoring of water distribution systems. Such an application can constantly monitor water quality parameters, with the possibility of sending warnings when an anomaly appears in the

system. The benefits of such a CPS can provide a balance regarding water quality or ensure consumer safety by detecting water-related issues [6].

Another water CPS application concerns the control and mitigation of water losses in a water distribution system. For example, in the United States, water losses are estimated at approximately USD 240,000/year. Therefore, such a problem can be solved by constantly monitoring the water pressure and possibly automatically closing the valves when a leak is detected [7]. Moreover, prediction models can calculate future values, making predictions and providing valuable recommendations [8].

To evaluate the impact of CPSs on existing solutions for water management processes, the following paragraphs emphasize the following two directions: (1) the applicability of cyber-physical systems in the field of water management, and (2) the benefits of blockchain technology in the context of cyber-physical systems.

An important application of CPSs for water management is related to decision support methods that provide forecasts, warnings, or recommendations. These can be essential in emergency situations, in order to be able to prevent danger or make good and quick decisions when they cannot be avoided [8].

In this scope, data acquisition from smart sensors and the processing of large data streams represent crucial components in the administration of water distribution systems based on IoT technologies. There are multiple challenges represented by the nature of the data streams which can require a different monitoring paradigm for large-scale operations. In this sense, the large number of rfds (relaxed functional dependencies) that may exist in a given dataset fluctuate continuously as new data are read from the stream. In paper [9], a visualization tool for rfds discovered in a data stream is described. The instrument permits the exploration of results for various categories of rfds and employs quantitative measures to track the evolution of discovery results. In addition, the application enables the comparison of rfds discovered in multiple iterations and provides visual manipulation operators for dynamically composing and filtering results.

The security part of cyber-physical systems is another important aspect when designing solid and robust architectures. Blockchain technology first appeared with the Bitcoin white paper [10], when the first cryptographic currency was born. Later, blockchain advanced, being used in many other fields, not only in crypto-currencies.

Blockchain is based on a decentralized peer-to-peer network. In this sense, there is not a single trusted entity, but any node in the network has the same rights as another participant of the consensus mechanism, increasing trust in the system.

CPS applications using blockchain can be divided into several directions. For example, one method by which blockchain can be used in CPSs is the tracking of employees and the objects they use. This aspect is useful in the case of post-incident investigations [11]. Many scientific papers detail the benefits of blockchain integration in CPSs from various industrial sectors [12–19].

Another field of activity in which blockchain has taken root is the medical one. In this sense, blockchain can be used in terms of medical research on patients' personal data [20]. Thus, the benefits are significant due to the impossibility of deleting or changing a record without leaving a digital trace to prove that this was done. Estonia and many other countries use this model to keep medical records secure. Another area in the medical field where blockchain is effective is drug tracking. There are CPSs that can help to scan medicines, ensuring that each one is added in secure digital blocks. Therefore, through this mechanism, all medicines have a trace/history, reducing the chances of counterfeiting. For extra protection, permissions can be added to give access to real-time recordings only to authorized parties [21].

Nowadays, the IoT has become indispensable, as it efficient and low cost. However, the combination of blockchain and the IoT is essential in the development of an advanced CPS based on emerging technologies. These technologies may offer us robust, complex, secure, and transparent solutions.

One such example is the Ethereum component of the blockchain that is used to communicate electricity consumption [22]. Through Ethereum, the network is notified when energy consumption exceeds a certain value and the devices are switched to economy mode, meaning they are used used efficiently.

A currently debated topic is smart homes. In paper [23], a solution is presented by which the general expenses of a house are significantly reduced. Moreover, a consortium of suppliers was proposed to facilitate the rapid trading of energy [24]. The collected information is used to create a network that can make decisions depending on the context.

Another blockchain approach in CPS applications is presented in [25], where Hyperledger Fabric plays an essential role by creating blocks, further used as smart contracts at the supervisor level. Furthermore, a CPS application that enforces access control policies via blockchain is presented in [26].

In the transport sector, autonomous vehicles have started to gain momentum in modern society. Their role is to make traffic more efficient by improving connectivity and road safety. Blockchain finds a role here as well, being used to create an intelligent, decentralized, secure, and robust CPS.

The intelligent refueling of autonomous electric vehicles using blockchain is presented in [27]. Furthermore, fast, reward-based communication is detailed in [28], enhancing privacy and improving communication between vehicles.

Crowdsensing leverages the ubiquity of mobile devices and the participation of individuals to collect and share data in real time. In the domain of water monitoring, crowdsensing has been used to gather data on water quality parameters such as temperature, pH levels, and pollutant concentrations. By engaging citizens as data collectors, crowdsensing enables large-scale and dynamic monitoring of water bodies, providing valuable insights into water quality variations and supporting early detection of potential hazards [2,4].

Participatory sensing platforms enable citizens to actively contribute to urban water management by providing data on water availability, usage patterns, and infrastructure conditions [29]. These platforms empower users to report issues, track water usage, and participate in decision-making processes related to water resource management [30]. As presented in [31], there are considerable advantages of crowdsensing solutions for real-time responses, as citizens can report observations in real time, helping authorities monitor flood-prone areas and improve the emergency response.

On the other hand, privacy and security considerations are essential in crowdsensing platforms to protect the confidentiality and integrity of user-contributed data. Studies have explored various techniques, such as data anonymization, access control, and secure data aggregation, to address these concerns and ensure the trustworthiness of crowd-sensed data [32–34].

Incentive mechanisms and gamification techniques have been applied in crowdsensing platforms to encourage citizen participation in water resource monitoring, e.g., collecting and sharing real-time data on water quality, availability, and usage and fostering a collaborative approach to water management [35–37]. By integrating game elements such as leaderboards, achievements, and rewards into data collection tasks, gamified applications can incentivize individuals to contribute to water monitoring efforts [38]. Several studies have shown the effectiveness of gamification in motivating participants and improving data quality in citizen science initiatives [39,40].

Simulations and virtual environments have been developed to simulate water management scenarios, allowing stakeholders to explore different strategies and their consequences in a risk-free environment. These gaming solutions have demonstrated the potential to enhance water conservation efforts and promote sustainable water use practices [41].

Furthermore, the integration with blockchain technology in crowdsensing scenarios has been shown to ensure data integrity, transparency, and trust among stakeholders. Blockchain effectively complements the incentive mechanisms involved in crowdsensing serious games with automated data validation, enhancing the overall reliability of crowdsensing solutions in large-scale monitoring processes [42–44].

To conclude this section, the major topics of interest in this research work were analyzed from an interconnected perspective. We started with cyber-physical systems and their application in multiple domains, with the primary focus on water management. We studied the cyber-physical solutions that were implemented in the water sector, the importance of these applications, and how they influence people's lives and health. Blockchain was analyzed in order to evaluate the current architectures and solutions that integrate this technology in multiple domains. Finally, the impact of crowdsensing and serious gaming on citizen involvement and the current implementations and contributions of these solutions in complex cyber-physical systems were studied.

## 3. Methodology

The aim of this chapter is to provide a theoretical context about cyber-physical systems in water management, encompassing emerging technologies and concepts, i.e., blockchain, crowdsensing, and serious gaming in the context of CPSs.

The research presented in Section 2 shows that there are many proposals in the water management area that support both water distribution flows and citizen involvement. On the other hand, the evolution of emerging technologies, e.g., blockchain, crowdsensing, and serious gaming, has had a major impact on the domain by allowing architects to design more complex and interactive solutions. However, there are still a few proposed solutions that integrate all these advanced technologies into a single architecture. Additionally, the studies demonstrate that the previously implemented architectures are based on conventional client–server architectures, in which only the providers/public institutions have direct access to and control of the information.

CPSs are complex systems that make the interaction between cyber and physical elements possible in an organized fashion, with steps that can be summarized as follows: (1) detection, (2) communication, (3) processing, and (4) control. As the internet changed the way people interact with information, CPSs propose to change the way people interact with systems, making them more sustainable.

As presented in Figure 1, CPSs are complex systems, conceptually based on three major elements: computing power, communication, and control [45]. This involves the integration of physical and digital elements (e.g., communication between sensors and other systems to transmit data that will be further processed). CPSs are characterized by real-time operation, scalability, and reliability, enabling modern systems to provide real-time monitoring, forecasting, and decision support capabilities.

From a structural perspective, CPSs can be defined as complex systems composed of three major components:

- A physical system (sensor/device);
- A network used to communicate/transfer data;
- A distributed cyber system.

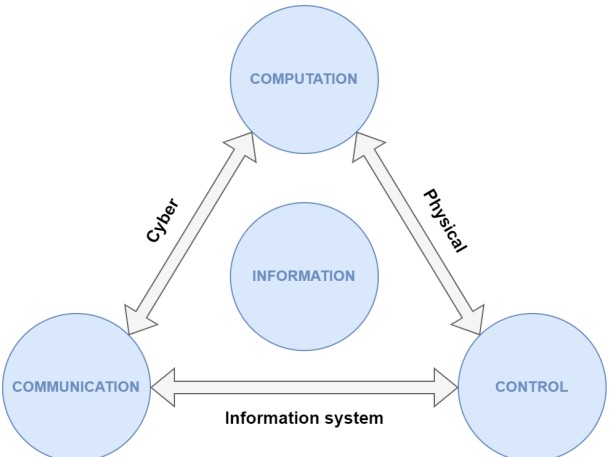

**Figure 1.** The 3C conception of CPSs [45].

Considering the context of CPSs as a central aspect of the proposed solution, Figure 2 depicts the solution overview, showing the main components defined in the context of urban water infrastructure management.

In this context, serious gaming represents an incentive-based approach based on crowdsensing, enabling gamified interactions for data collection, analysis, and stakeholder engagement. Serious games have been shown to engage citizens in water quality monitoring initiatives, facilitating data collection and promoting environmental awareness. These games often incorporate educational elements, interactive interfaces, and real-time data visualization, allowing users to actively participate in urban water management [5,46,47].

From the blockchain perspective, the Ethereum-distributed ledger enables the creation and execution of smart contracts that provide support for transparent, secure, and decentralized interactions and contract management between stakeholders.

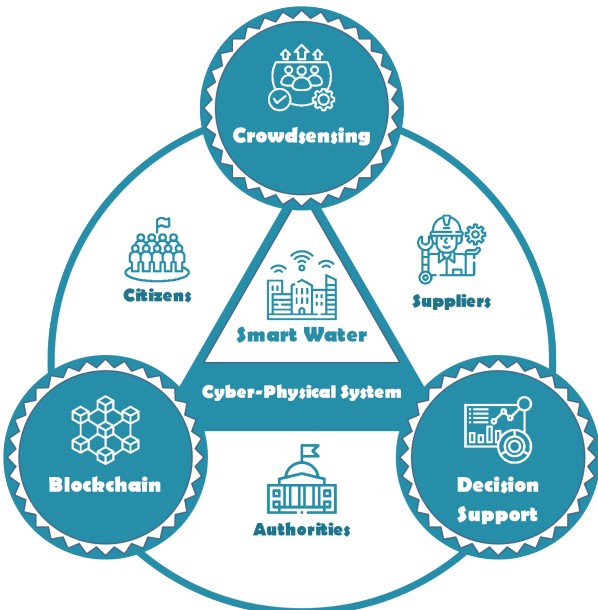

**Figure 2.** Solution overview.

Since its introduction, Ethereum has become the most important platform for developing decentralized applications (dApps) and issuing digital tokens. From a technical point of view, the Ethereum architecture consists of the following:

- A chain of blocks that contain lists of transactions;
- The Ethereum virtual machine, which is the runtime environment (as a secure sandbox) that executes smart contracts written in Solidity;
- The proof of stake algorithm and the native ETH token which is used both for transactions (regardless of type) and fees [48].

On the other hand, Ethereum-specific tokens, i.e., ERC-20 [49] and ERC-1155/ERC-721 [50], bring a lot of possibilities for developing new architectures. The key features that come with Ethereum ERC-20 can be summarized as follows:

- A fixed supply (that makes the coin deflationary and stable);
- The possibility to transfer and interrogate account balances on demand in a very simple manner;
- Events that can be easily tracked in a decentralized architecture.

Moreover, ERC-1155 (multi-token contract) comes with the possibility to manage multiple token types in a single contract, being able to handle both fungible and non-fungible assets, transfer these assets in batches, and also manage metadata (or URIs) for the owned tokens [49,50].

All these technical aspects and ready-to-use features allowed us to use the Ethereum blockchain as the primary framework for implementing a decentralized urban water infrastructure management solution in the context of this work.

## 4. Baseline Solution

In this section, the baseline solution is presented. The presented work is based on previous research and aims to continue and extend the proposed solution. The initial architecture consists of a crowdsensing platform with gamified interactions in the context of a water distribution reporting application.

The baseline solution for event reporting implements crowdsensing scenarios in the urban water infrastructure. The reporting system integrates a crowdsensing model into a serious-game-based interactive platform with blockchain support.

Considering the location-based serious game architecture, where reporting is distributed across the map, the incentive mechanism is designed to achieve optimal player distribution for collaborative issue reporting in the crowdsensing context. This problem can be defined as an allocation optimization problem combined with a reward-based incentive mechanism. Tasks involve reporting issues in the context of the urban water infrastructure, as described in previous research [35].

The application developed within the Watergame project (mentioned in the funding section) is available in web format, adapted for mobile devices, and includes the functionalities for the urban water infrastructure event reporting solution. The validation scheme is based on reported issues that can be uploaded and confirmed by other users based on their assigned roles: Finder (reporting events), Fixer (resolving reports), and Validator (validating reports).

The event reporting interface is based on Google Maps and the nearby reports can be visualized and represented as markers, as can be seen in Figure 3. Each report is recorded in the database with a unique identifier. Information about reports can be viewed publicly, and the application allows the validation, resolution, update, or deletion of a report. The user's location on the map is determined by the location provided by the browser and is represented by the blue marker for the Finder user (a yellow or red marker for the Fixer or Validator users). Using the map interface, users can add new reports interactively or view reports uploaded by other users.

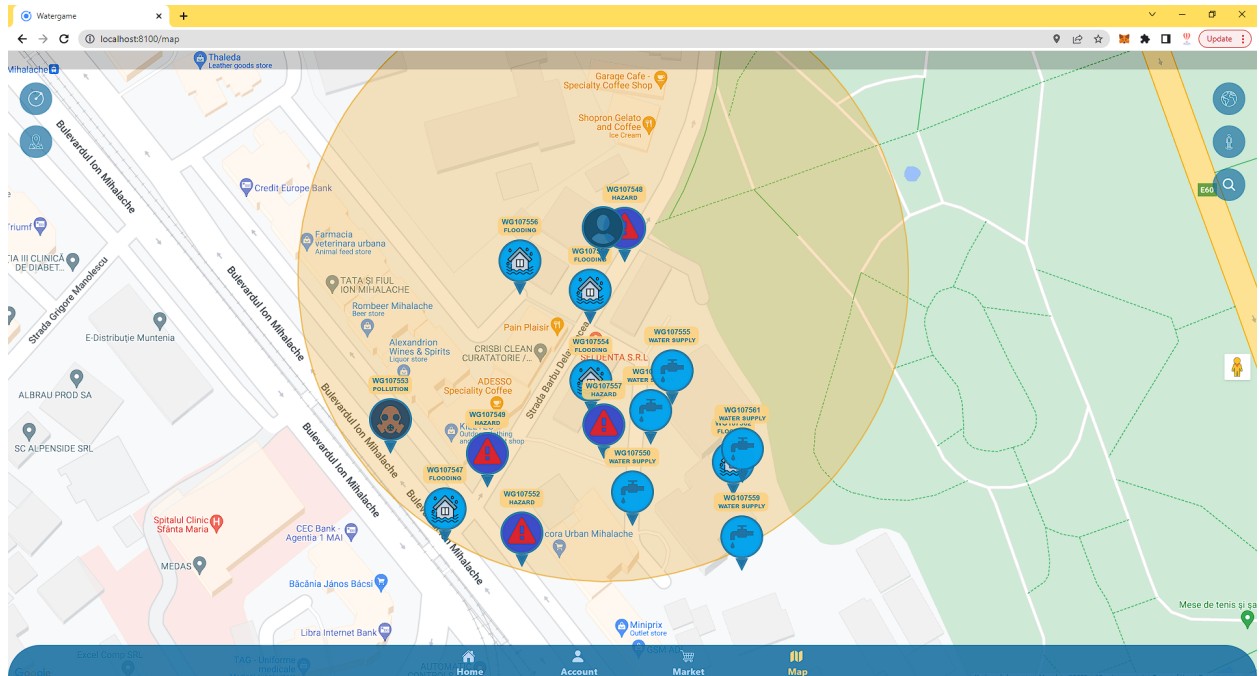

**Figure 3.** Interactive reporting.

The user's profile in the app is shown in Figure 4, and consists of a visual representation of the user's progress in the serious game, according to the proposed crowdsensing roles. Based on this profile, the user's reputation is calculated, which calculated by the weight of their actions in the interactive reward process. To increase the overall utility of the platform, the reward-based incentive is counter balanced by a regulatory strategy. Reputation is the method of validating participants and is defined to achieve optimal incentives in reporting activities while reducing the risk of over-reporting or false reporting. The reputation is determined by user interactions in the context of urban water infrastructure event reporting scenarios.

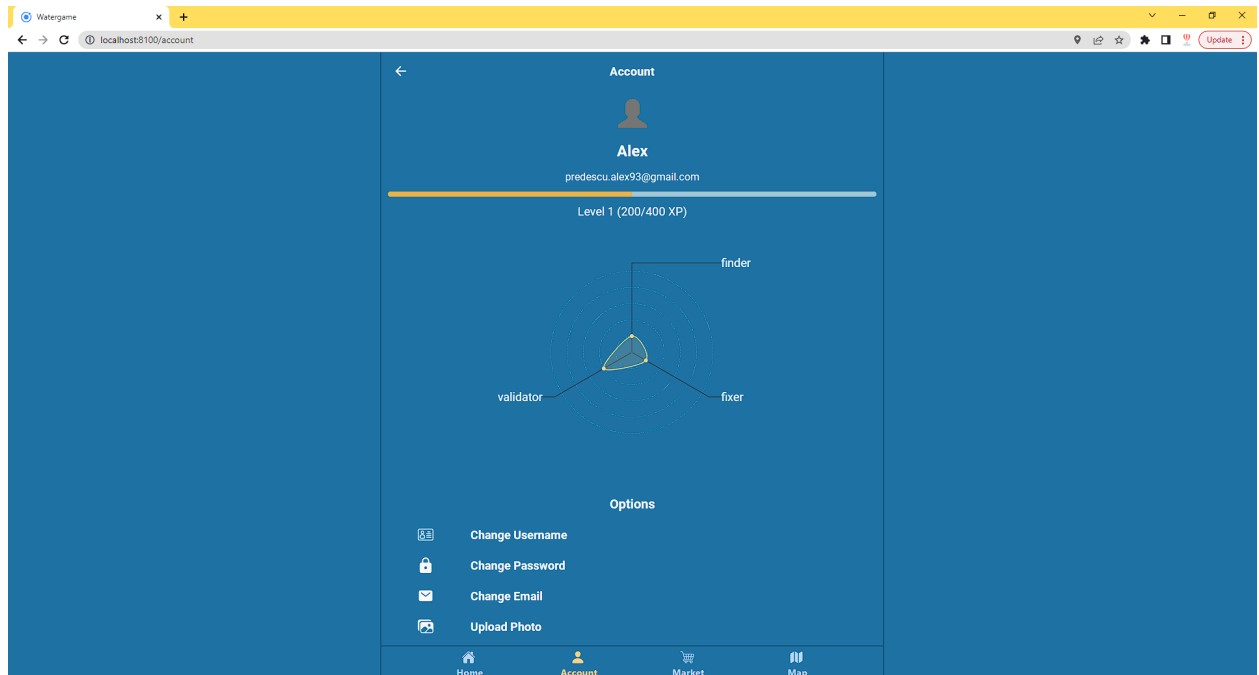

**Figure 4.** Crowdsensing user profile.

When users interact, related information is recorded as follows: a user can record the validation of a report only once, while there can be multiple validations from different users. Considering the technical infrastructure, the architecture includes a relational database, used to store and retrieve the reports and their associated data. Considering the complexity of the integrated system in large-scale deployments, scalability is an important aspect. At the service layer, extensions were added to improve fault tolerance, while a cache subsystem was added to increase the throughput of database data processing. The database was designed to handle complexity by indexing key attributes and optimizing queries, while data transfer was optimized by using local caching to reduce the amount of data that are transferred between the web application and the server.

## 5. Proposed Solution

After the introduction of Ethereum, together with smart contracts, the adoption of blockchain technologies increased in a significant way, with the ecosystem providing a strong and stable environment to develop decentralized solutions. Based on the Ethereum blockchain, other projects were developed, each with a new vision of what the software architecture should look like [51].

Starting from the baseline solution, which was designed using the traditional architecture, the proposed architecture is based on the latest blockchain features with a decentralized approach, web 3.0 concepts, and token standards such as ERC-20 and ERC-1155 [49,50].



One of the aims of this paper was to tokenize the user's rewards and migrate from saving the rewards in a centralized cluster (relational database) to using the ERC-20 token (decentralized architecture on the blockchain). Figure 5 shows what the solution architecture looks like.

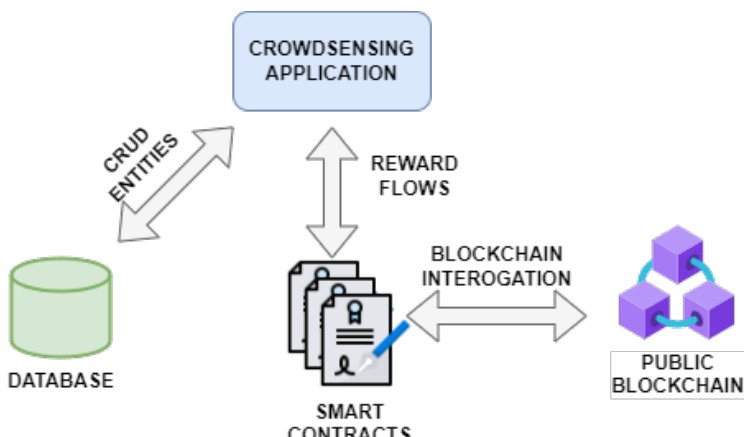

**Figure 5.** Initial architecture.

First of all, from a design point of view, the linking point in the proposed solution is the newly generated ERC-20 token that will serve as a reward. ERC-20 tokens are smart contracts (i.e., a piece of code) that are deployed on the decentralized network and facilitate all the operations that will be performed on the application. As mentioned before, the token will be offered as a reward after reporting a valid incident in the urban water infrastructure [49].

On the application level, there are several other ways in which this ERC-20 token can be obtained:

- Donation (from another user);
- Participation in an ICO (on the application level an initial coin offer for the ones that want to buy this token before solution release);
- Buying/trading (directly from a crypto-wallet, exchanges, or other platforms).

One of the main benefits of this approach is that the token can be traded outside of the application using any crypto-wallet, directly from any laptop or phone, without affecting the application functionality. In addition to this ERC-20 contract, some other contracts that facilitate the previously presented actions are the Rewarder Contract, Donor Contract, and TokenSale Contract.

- Rewarder Contract
    - Basically used as an admin contract for all the tokens that will be provided as a reward for users.
    - The contract will have a limited amount of tokens to administrate, but will be a mintable-like contract (there will be the possibility to add more tokens on the existing supply on demand).
    - The contract will have the possibility to pause/start the rewarding on request only by admin if there is any critical issue that requires stopping the reward.
- Donor Contract
    - Basically used as an admin contract for all the tokens that will be provided as donations between users.
    - The contract is just an extra layer of the ERC-20 token contract that allows passing tokens from one account to another, with the sender paying the transaction fees.
    - The contract will have the possibility to pause/start the donations on request only by admin if there is any critical issue that requires stopping the donations.

- TokenSale Contract
    - This contract is the one that is used in order to facilitate the Initial Coin Offer mechanism.
    - The contract will have a limited amount of tokens to administrate, but will be a mintable-like contract (there will be the possibility to add more tokens on the existing supply on demand.)
    - The contract will have the possibility to pause/start the token sale on request only by admin if there is any critical issue that requires stopping the donations.

The main scope of the proposed solution/extension is to digitalize the items, discounts, or any other benefits that can be purchased with the token received as a reward. For this approach, the best option is the ERC-1155 token standard.

The role of the discount in the baseline solution is to provide the user with the possibility to obtain price reductions based on their interaction with the application. In the new proposed architecture, the discounts will be stored in a decentralized way on blockchain as an ERC-1155 token. The main benefit of this approach is that the asset can be easily transmitted from one person to another directly from a wallet or on a secondary market. By doing this, the discount applied to the prices is directly transferred without any additional operations from the application administrators. From a technical point of view, there are two contracts that handle this functionality:

- Discount Contract
    - ERC-1155 Contract.
    - The contract will be mintable, with the possibility to add more tokens if needed.
- Marketplace Contract
    - Custom contract to facilitate the discount buying process from the marketplace application.
    - Basically a wrapper over the Discount Contract.
    - The contract will have the possibility to pause/start the sale on request only by admin if there is any critical issue that requires this action.

As a flow diagram, the actions that can be supported based on the new architecture are presented in Figure 6.

According to the diagram presented in Figure 6, the process starts inside the crowdsensing application by reporting an incident in the water distribution network. After reporting, the incident is saved in the application's relational database, and the reporter receives a reward that consists of a digital token. The process of receiving tokens involves interaction with the blockchain to transfer tokens from the application administrator wallet to the client wallet. Independently, the user is able to buy, trade, or donate tokens as they want from their wallet (this action also implies interaction with blockchain smart contracts). Once the client has tokens in their wallet, they are able to purchase contract subscription discounts. The user will access the marketplace section and buy the discount. An ERC-1155 token kept on the blockchain will serve as the discount's representation. Basically, the user will trade native WGT tokens for a discount token. Once the discount token is purchased, the physical discount will be automatically saved in the relational database without any interaction from the solution administrator. The system will permanently scan for users' digital discounts and update the physical ones in the relational database. Separately from the application, by using blockchain, the user will be able to buy, sell, and trade both application-native tokens and digital discounts on other trading platforms.

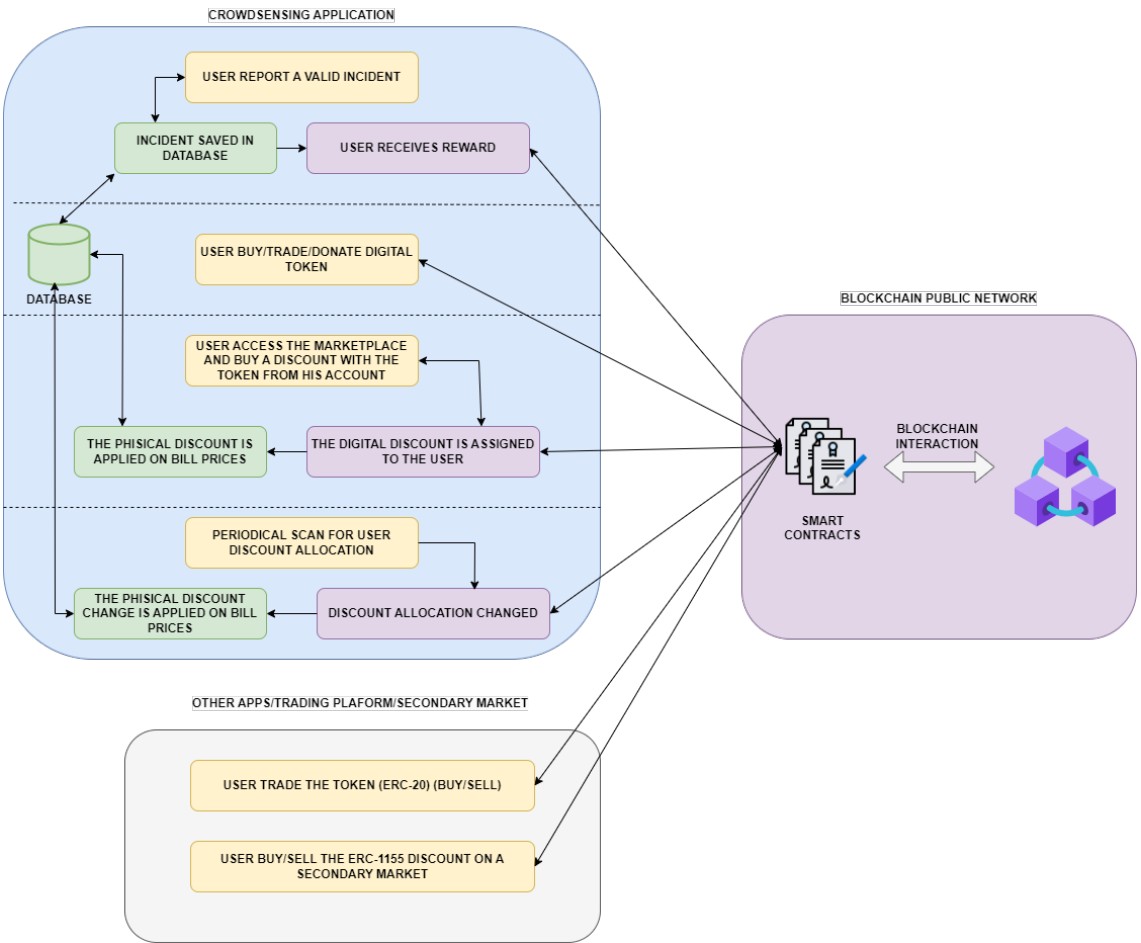

**Figure 6.** Solution design on the decentralized architecture.

## 6. Results

The main results of this paper consist of designing a decentralized application in which clients can report incidents in a water distribution network, obtain rewards after the reporting, and purchase subscription discounts, all by using a decentralized solution based on Ethereum blockchain.

Figure 7 shows the integrated blockchain interface using the MetaMask extension, which allows performing operations with the implemented WGT (Water Game Token) virtual currency.

Upon authentication on the blockchain network, the user account information (account identifier and balance) is retrieved and displayed in the application. After that, the clients will be able to see lists of the discounts available for purchase using the WGT virtual currency.

The WGT virtual currency can be obtained from the activities and user interactions with the application and is assimilated into the rewards that can be granted according to the proposed crowdsensing scenarios. The purchase of discounts from the catalog involves the transfer of an amount of virtual currency from the user/consumer account to the supplier account, after which the product will be registered on the user account.

By implementing this functionality, the user will be able to purchase subscription discounts that will be automatically applied to their account. Moreover, because the discounts are saved as ERC-1155 tokens, whenever the user sells their discount (on the secondary market or other marketplaces outside of the application), the subscriptions will be automatically changed and the discount will not be applied.

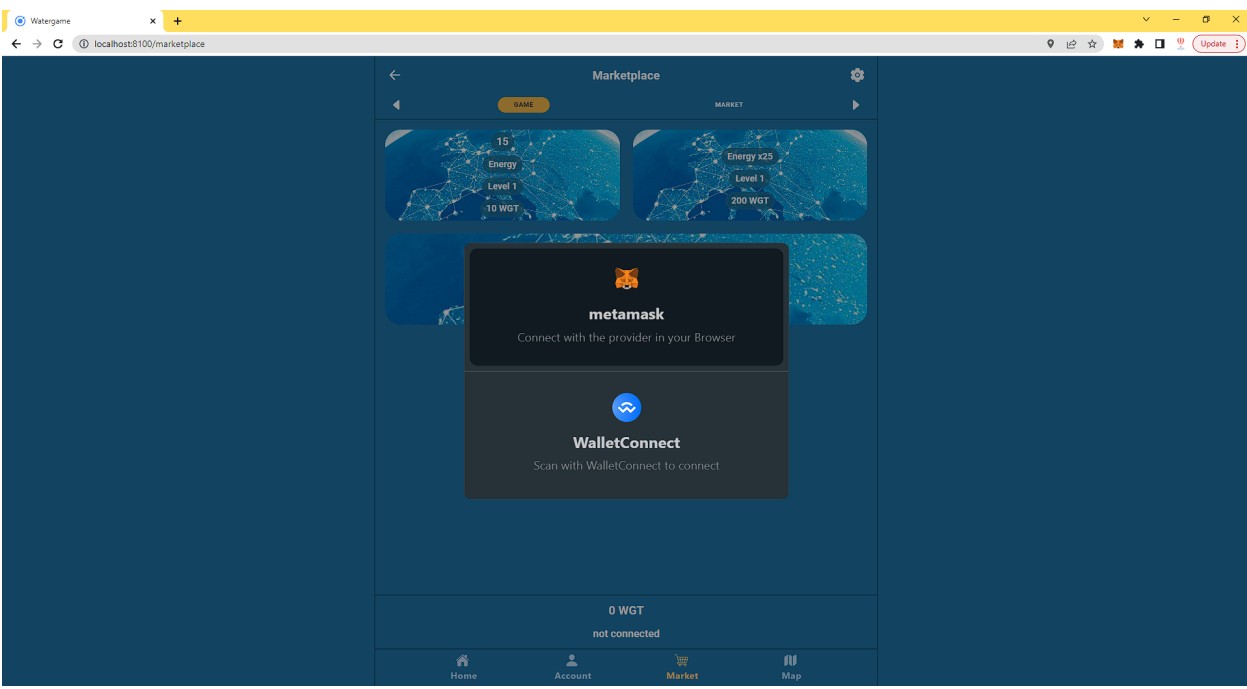

**Figure 7.** Metamask interface.

Figure 8 presents how the application will work on a hybrid architecture that uses both centralized and decentralized components.

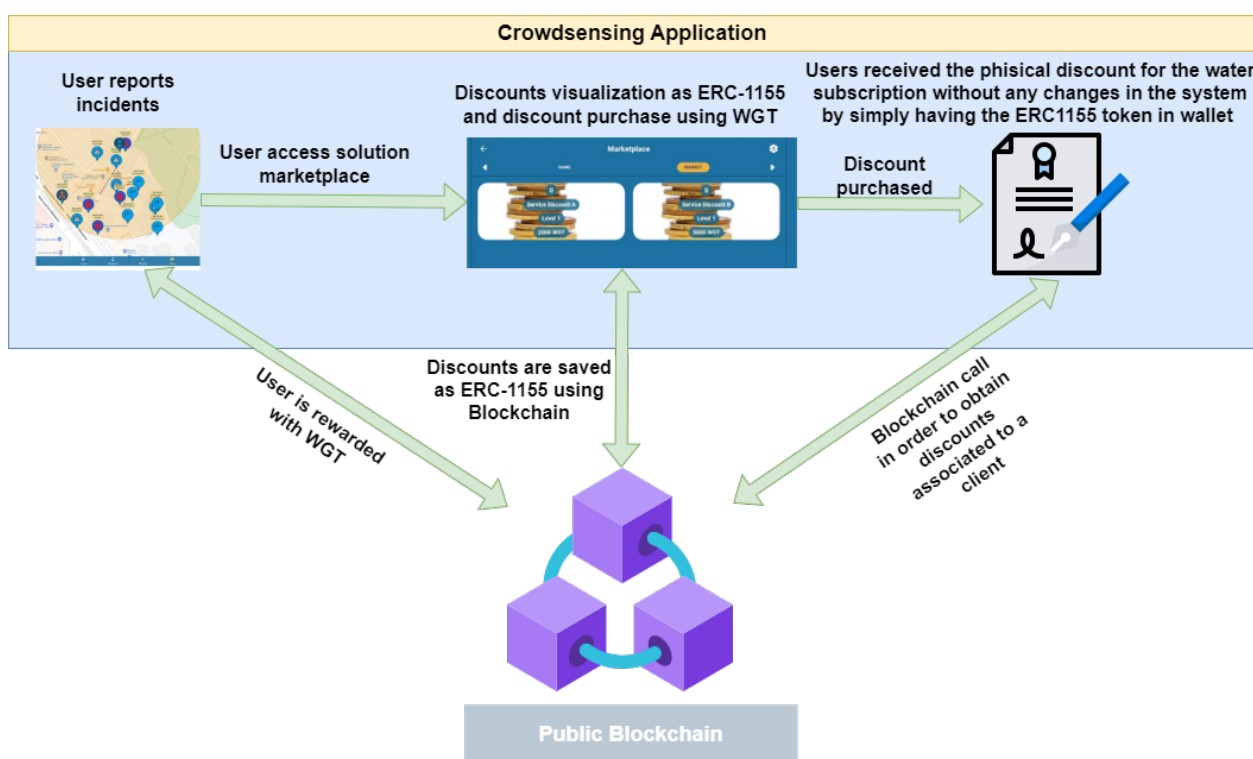

**Figure 8.** Hybrid application flow.

As presented in the figure, the user reports incidents in the water distribution network and is rewarded with WGTs (Water Game Tokens). In order to buy physical discounts on their water subscription contract, the user accesses the application marketplace. On the marketplace page, the user sees all the available discounts and the price of these discounts.

The user selects a discount and buys it in exchange for WGTs. After the purchase is finished and the transaction is inserted on the blockchain level, users receive the virtual discount (ERC-1155) and also the physical contract price reduction.

In order to maintain the ERC-1155 token implementation, a decentralized method to store the discount images and metadata was required. For managing the discount tokens, the following decentralized file system was used: https://www.pinata.cloud/ (accessed on 12 May 2023). Figure 9 presents the way in which the ERC-1155 token is saved on the file system.

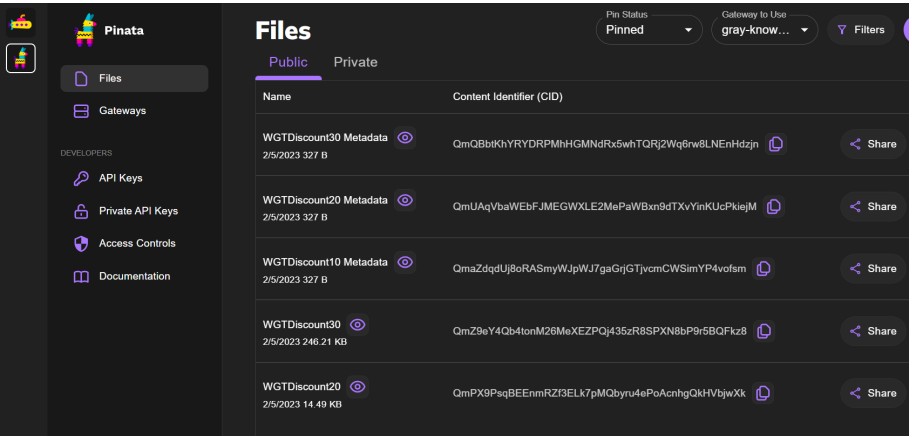

**Figure 9.** ERC-1155 Pinata visualization.

On the IPFS level, for each discount that can be given inside the application, there were two files that were created: a metadata file and a discount image. The metadata file is used to define the virtual discount and also to obtain important information like discount value and discount image.

The discounts are saved using some metadata files that contain the following properties:

- Discount name (e.g., WGTDiscount30);
- Description (e.g., 30% WGT discount on water contract);
- Token image URL;
- Discount value (e.g., 30);

In order to evaluate the implemented solution, two approaches were compared: the initial centralized architecture and the new one based on decentralized components. The novelty of the new approach was the introduction of blockchain technology in this architecture.

By adding this component, the application security and trust were increased while providing a way for end users to monetize their rewards in the real world. Therefore, each user will be able to use their tokens or discounts inside the application or simply sell them on public exchanges. Another important aspect that needs to be underlined is the fact that all the operations are transparent and represented as blockchain transactions.

Last, but not least, a lot of paperwork can be eliminated by automatically applying the subscription discounts.

The downside of implementing this approach is determined by the fact that the application complexity significantly increases by using blockchain. The complexity comes from both a technical point of view (related to the communication and integration between centralized and decentralized components) and from a user point of view, by adding the requirement of using a crypto-wallet, i.e., Metamask.

From a technical point of view, even though the solution complexity was increased, the implemented hybrid architecture fits the needs in the presented context and provides the end user a better experience and more advantages in using the solution on its true value.

## 7. Conclusions

The aim of this paper was to define the migration of an existing cyber-physical water management solution from a traditional architecture using a relational database to a hybrid model using both centralized and decentralized components that are easily interconnected. The solution design was created after extensive research regarding both the technologies used in the base solution and also the possible benefits that blockchain can bring. The research shows that blockchain may be a game changer in our architecture and may offer a new vision for both water providers and clients.

Our proposal brings novelty to the sector by combining concepts such as crowdsensing and serious gaming using their specific characteristics, and most importantly, applying blockchain on top of the architecture in order to increase solution security, transparency, and trust. Another important aspect that brings novelty to our paper is the fact that by using blockchain, users are able to manage their assets transparently, e.g., rewarding tokens and trading NFT discounts.

The starting point was creating an ERC-20 token written on the Ethereum blockchain and using it as a rewarding mechanism for incident reporting in the urban water infrastructure. Based on this token, the second iteration of the implementation was to define and implement the decentralized mechanism for purchasing and using the discounts applicable to user subscriptions. This component is fully decentralized; thus, all the assets (ERC-20—WGT Token or ERC-1155—Reward token) can be also traded outside of the implemented solution using a cryptocurrency wallet.

From a functionality point of view, the rewarding process was moved from a centralized to a decentralized architecture using blockchain. This new approach offers the user the possibility to use the received digital token both within the integrated solution and outside it, i.e., on public exchanges or other crypto platforms. Furthermore, by using the WGT token, customers may be able to buy contract discounts from suppliers. Similarly, the novelty of our proposal is determined in this case by the fact that all the discounts will be applied directly based on the smart contract. Moreover, if the client wants to sell their discount, the subscription will be automatically updated and no additional intervention will be required from the supplier's point of view. Last but not least, the token price increases that usually occur during usage of the blockchain solution will make it possible for customers to buy and sell app tokens both inside and outside the application and make a profit.

In conclusion, it can be stated that with the new version of our application, a big step forward was taken and major improvements in security, openness, trust, and user engagement areas were added.

As the next steps for the existing implementation, the focus will be on creating a decision support system, also based on the ERC-20 WGT token, that will facilitate decision making from the water provider's point of view.

**Author Contributions:** Conceptualization, B.-I.P., A.P. (Alexandru Predescu); Software, B.-I.P., A.P. (Alexandru Predescu); Writing—original draft, B.-I.P., A.P. (Alexandru Predescu) and D.A.A.; Writing—review & editing, A.P. (Adrian Petcu); Supervision, M.M. All authors have read and agreed to the published version of the manuscript.

**Funding:** This research was funded by UEFISCDI, grant number PN-III-P2-2.1-PED-2019-4993, Smart Urban Water Based on Community Participation Through Gamification–Watergame Project. The results presented in this article have been funded by the Ministry of Investments and European Projects through the Human Capital Sectoral Operational Program 2014–2020, Contract no. 62461/03.06.2022, SMIS code 153735. The publication was supported by the National University of Science and Technology Politehnica Bucharest through the PubArt program.

**Data Availability Statement:** Not applicable.

**Conflicts of Interest:** The authors declare no conflict of interest.

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
