# Peer review of "A Blockchain Approach for Migrating a Cyber-Physical Water Monitoring Solution to a Decentralized Architecture"

_water, doi:10.3390/w15162874_

Round 1

Reviewer 1 Report (New Reviewer)

General comment:

This paper study about the actual state of the existing architectures implemented in the water management field with support for blockchain technologies. Overall, the paper focus more on the development and implementation of an application. The paper seems like fell out of the scope of water related aim. The results were poorly explained with limited justification to support the claim. The overall structure also needs to be improved to let the reader understand. To conclude, this paper needs to revise and restructure the entire manuscript carefully before it can be considered in impact journal. Hope below comments will able to help to further improve the paper.

Specific comment:

Abstract:

-      Article should be written in passive manner. Avoid using ‘we’ when writing.

-      Suggest to modify the title to be more attractive and related to the current trends.

-      Needs major revision prior to the amendment of the main content.

-      An abstract is often presented separately from the article, so it must be able to stand alone. Hence the problem statement, aim, novelty and results of the study, all should be included into the one paragraph of abstract.

Introduction:

-      Introduction should be written in several paragraph without any subsections. Please revise the structure of the introduction.

-      Introduction should be covered the gap of the research. However, it is not well covered in this section.

-      Also, please mention the important of this study to society as well as industry.

-      Problem statement of your introduction is not strong, need to discuss more about it.

-      A good introduction should conclude the introduction by mentioning the specific objectives of the research. The earlier paragraphs should lead logically to specific objectives of the study.

-      Revised Introduction section based on the structure below:

1st paragraph: Problem statement

2nd paragraph: Current ongoing solution

3rd paragraph: Proposed solution in this work.

4th paragraph: Summarized the current research novelty and objective of this work.

Related work:

-      Please move the aim and objective to the introduction section.

-      Article should be written in passive manner. Avoid using ‘we’ when writing.

-      Line 169; ‘… cite2011StateOfTheArt.’ What is this statement? Please revise.

-      What is the main purpose of the section? Why is it name as related work as its not case studies related to the work or so on. Is more on explaining the terms.

-      Why is there a conclusion in the section?

-      Too lengthy and hard to understand. Please revise to more straight to the point and highlight the key points or importance.

-      Please merge all the small paragraphs. The structure needs to be improved.

Baseline Solution & proposed solution:

-      The heading was name as baseline solution but line 306 wrote proposed solution?

-      What is the different in baseline and proposed solution? Isn’t it just a continuation?

-      The way the author writing the proposed solution was confusing and hard to understand. Please revise.

-      Many statements were not supported by reference which the reliability was questionable.

-      What is the duration of the experiment?

Results:

-      Article should be written in passive manner. Avoid using ‘we’ when writing.

-      The results were hard to understand. What is the main result?

-      The figures and images showed in the paper is in low resolution. Please provide higher quality of figures and images.

-      The overall structure needs to be improved.

-      Kindly improve on the discussion. What is the significance of the results of the work?

-      There's no statistical analysis in some data. It is always good to have some statistical analysis in the results as it will strengthen the discussion.

-      Authors should pay attention to English grammar and sentence structure. The authors are suggested to edit the manuscript carefully before submission.

Conclusions

-      It is suggested to include additional information or clarifications to the methods and results sections to evaluate the manuscript's novelty and its significance to the field.

-      Kindly improve to more concise with significant results.

References

-      Kindly revise reference format according to the author guideline.

English very difficult to understand/incomprehensible

Author Response

Thank you for your detailed review. We appreciate your valuable suggestions, which contributed greatly to improving the quality of our paper. We revised the entire paper and improved the overall structure as outlined by the following changes.

Reviewer 2 Report (Previous Reviewer 2)

Thanks to the authors for following the previous comments. I don't have other comments to raise.

Author Response

Thank you for your review. We appreciate your work, and we are glad that the previous comments were fulfilled. We did some additional restructuring of the paper (according to other reviewers' suggestions) in order to offer more clarity to our research.

Reviewer 3 Report (New Reviewer)

This manuscript uses blockchain technology in a water distribution management clock, facilitating the migration of functions in the management system from a centralized to a decentralized architecture in a public distribution network, and describes the actual implementation and the challenges and solutions found. This paper has good application value and relevance. The manuscript is well organized and easy to understand.Just some remarks:

1.The results and insights are case specified. This is fine for a research project, but cannot be considered as academic contributions.

2. Try to highlight the strengths and limitations of the work more clearly.

3. Renegading the aspect just mentioned, literature review part may also be strengthened.

The written English must be improved.

Author Response

Thank you for your review. We revised the entire paper according to the reviewers’ observations and improved the overall structure as outlined by the following changes.

Round 2

Reviewer 1 Report (New Reviewer)

The manuscript is corrected and revised according to the reviewer's comments. I am now satisfied with the new version, so I would like to recommend its publication.

Minor editing of English language required

This manuscript is a resubmission of an earlier submission. The following is a list of the peer review reports and author responses from that submission.

Round 1

Reviewer 1 Report

My major concern about this manuscript is the lack of any practical results . The advances smart technologies are shortly described which could be the goal of a short review paper or technical communication. The MDPI journal Water (as indicated by its aims  ans scopes) is strongly related to problems of water science and technology and the present manuscript does not contribute significantly in these fields.

Of course, the information presented is interesting and important and deserves publication but in a MDPI journal dedicated to computer technologies.

The English language needs minor improvements

Reviewer 2 Report

Research summary: 

The topic is relevant and may be of interest to a broad range of the journal's readers. However, this reviewer has some major concerns about the paper.

Major Strengths: The major strengths of the research are:

- The topic is interesting

Major Weaknesses: The major weaknesses of the research are:

- The structure and contents of the paper need to be improved.

- The proposed approach seems to be simple

Grammar and Readability:

The paper requires a revision of the English, and some parts are unclear.

Specific Comments: My specific comments concerning this manuscript are:

- The abstract does not highlight the specifics of the research or findings but contains too much background information. Some details of the research would be nice for example numbers addressing the sample, data, percentage improvement, etc.. Remove some of the background material and add some details of the research. Moreover, it is good to provide some specifics (e.g., sample size, dataset size, numbers from results, etc.).   

- Although a novel combination might be allowed, it is necessary to highlight the contribution of such a combination from both methodological and empirical perspectives. Also, it is required to provide technical details of the proposed methods as much as possible and in-depth explanations of method selections. 

- The innovation of the paper seems limited. The proposed method is a straightforward combination of existing techniques, which makes it less innovative. Also, more details about the proposed method should be provided.

- There are several papers that have addressed similar problems, but it is necessary to further highlight the novelty between the proposed study and the related literature.

- Starting from the previous works, I suggest introducing a table to summarize the most recent works and to highlight the novelty of the proposed work.

- Please add more recent references. Certainly, there has been more recent (within the last two years) research on this topic published in information science and/or computer science outlets. An academic search on the topic (using keywords from the manuscript’s title) shows that there is recent work in this area. Therefore, authors must update their literature review. As an example, some related works that have not been treated are those related to interactive interfaces, although they seem to be correlated. I suggest some of them: https://doi.org/10.1016/j.bdr.2021.100240, https://doi.org/10.1002/admt.202200757, https://doi.org/10.1002/admt.202200757

- There needs to be an explicit research objective(s) and/or research question(s) stated, preferably as a separate section. This helps readers find out what the research is trying to address.

- The reference list needs tidying up, as there are references missing items or formatting issues. Please be consistent with the formatting and use some standard formatting style. 

- The paper lacks a consistent evaluation. Please consider using a more convincing way to evaluate the proposed method.

- Check the references that are not linked in the paper, i.e., [?]

- The discussion of the results does not highlight the strengths and weaknesses of the proposed approach.

- Snippet of code and storing format should be removed.

- 2.6. Conclusion is an empty subsection 

Concluding Remarks:

I think that the paper could be improved with the considerations I reported in the review, but this version is not ready for publication.